# Severe Acute Kidney Injury in Cardiovascular Surgery: Thrombotic Microangiopathy as a Differential Diagnosis to Ischemia Reperfusion Injury. A Retrospective Study

**DOI:** 10.3390/jcm9092900

**Published:** 2020-09-08

**Authors:** Melissa Grigorescu, Christine-Elena Kamla, Dietmar Wassilowsky, Dominik Joskowiak, Sven Peterss, Stephan Kemmner, Maximilian Pichlmaier, Christian Hagl, Michael Fischereder, Ulf Schönermarck

**Affiliations:** 1Division of Nephrology, Department of Medicine IV, University Hospital, LMU Munich, D-81377 Munich, Germany; Michael.Fischereder@med.uni-muenchen.de; 2Department of Cardiac Surgery, University Hospital, LMU Munich, D-81377 Munich, Germany; Dominik.Joskowiak@med.uni-muenchen.de (D.J.); Sven.Peterss@med.uni-muenchen.de (S.P.); Maximilian.Pichlmaier@med.uni-muenchen.de (M.P.); Christian.Hagl@med.uni-muenchen.de (C.H.); 3Department of Anaesthesiology, University Hospital, LMU Munich, D-81377 Munich, Germany; Dietmar.Wassilowsky@med.uni-muenchen.de; 4Transplant Center, University Hospital, LMU Munich, D-81377 Munich, Germany; Stephan.Kemmner@med.uni-muenchen.de

**Keywords:** acute kidney injury, cardiovascular surgery, aortic aneurysm, aortic replacement, thrombotic microangiopathy, atypical hemolytic uremic syndrome, plasma exchange, eculizumab

## Abstract

Background: Acute kidney injury (AKI) after cardiovascular surgery (CVS) infers high morbidity and mortality and may be caused by thrombotic microangiopathy (TMA). This study aimed to assess incidence, risk factors, kidney function, and mortality of patients with a postoperative TMA as possible cause of severe AKI following cardiovascular surgery. Methods: We analyzed retrospectively all patients admitted to the ICU after a cardiovascular procedure between 01/2018 and 03/2019 with severe AKI and need for renal replacement therapy (RRT). TMA was defined as post-surgery-AKI including need for RRT, hemolytic anemia, and thrombocytopenia. TMA patients were compared to patients with AKI requiring RRT without TMA. Results: Out of 893 patients, 69 (7.7%) needed RRT within one week after surgery due to severe AKI. Among those, 15 (21.7%) fulfilled TMA criteria. Aortic surgery suggested an increased risk for TMA (9/15 (60.0%) vs. 7/54 (31.5%), OR 3.26, CI 1.0013-10.64). Ten TMA patients required plasmapheresis and/or eculizumab, and five recovered spontaneously. Preoperative kidney function was significantly better in TMA patients than in controls (eGFR 92 vs. 60.5 mL/min, *p* = 0.004). However, postoperative TMA resulted in a more pronounced GFR loss (ΔeGFR −54 vs. −17 mL/min, *p* = 0.062). There were no deaths in the TMA group. Conclusion: Our findings suggest TMA as an important differential diagnosis of severe AKI following cardiovascular surgery, which may be triggered by aortic surgery. Therefore, early diagnosis and timely treatment of TMA could reduce kidney damage and improve mortality of AKI following cardiovascular surgery, which should be further investigated.

## 1. Introduction

Acute kidney injury (AKI) after cardiovascular surgery (CVS) is a serious complication associated with high morbidity and mortality [1,2], resulting in prolonged length of in-hospital and ICU stay and a three- to eight-fold higher mortality [1]. Patients with cardiovascular surgery-associated AKI (CSA-AKI) have an increased risk for developing chronic kidney disease and end-stage kidney disease (ESKD) [1,3]. The incidence of CSA-AKI varies widely, ranging from 4% to 42% [3]. AKI severity may range from asymptomatic rise of creatinine to the need of renal replacement therapy (RRT), the latter occurring in approximately 1–5% of patients with CSA-AKI [1,3]. Preoperative risk factors for CSA-AKI include female sex, advanced age and multiple comorbidities such as preexisting chronic kidney disease, previous cardiac surgery and other cardiovascular risk factors [3]. The pathophysiology underlying CSA-AKI is likely multifactorial. Ischemia-reperfusion injury, renal hypoperfusion, mechanical factors and the use of nephrotoxic agents have received most attention [1,4]. Additionally, intraoperative use of cardiopulmonary bypass (CPB) causing hemolysis, bleeding complications and low cardiac output can contribute to renal hypoperfusion [1].

Thrombotic microangiopathies (TMA) represent a complex of very rare diseases characterized by platelet consumption, microangiopathic hemolytic anemia (MAHA) and organ dysfunction resulting from endothelial damage and microvascular thrombosis [5]. They are classified in three different disease entities according to the underlying pathophysiology: thrombotic thrombocytopenic purpura (TTP), caused by severe deficiency of the von-Willebrand Factor-cleaving protease called ADAMTS13 [5], hemolytic uremic syndrome (HUS), caused by Shiga toxin-producing *Escherichia coli* (STEC-HUS) and atypical HUS (aHUS) [6]. The latter is the historically designated term for all phenotypical TMA presentations upon exclusion of TTP and STEC-HUS [7]. In general, clinical presentation involves a degree of renal impairment and other extrarenal manifestations such as neurological impairment (impaired consciousness, confusion, drowsiness, seizures), gastrointestinal symptoms and cardiac abnormalities [5].

Atypical HUS can be classified as primary and secondary. Primary aHUS (“complement-mediated aHUS”) results from the uncontrolled activation of the alternative complement pathway at the endothelial cell surface [8]. Secondary aHUS or trigger-induced aHUS refers to clinical conditions associated with the aHUS-phenotype such as malignant hypertension, renal disease, pregnancy, infections, chemotherapy, cobalamin-c-deficiency, malignancies, autoimmune diseases, and solid-organ transplantation. Many clinicians refer to this phenomenon as secondary TMA or TMA with associated triggers [9]. Treatment of primary aHUS consists of eculizumab, a humanized monoclonal antibody against C5. However, its use in secondary aHUS/TMA forms remains controversial [6].

Various studies concerning surgical interventions not related to transplantation have reported the development of TMA in the postoperative setting procedures [9]. Three case reports describe the new onset of aHUS as a differential diagnosis to AKI following cardiovascular procedures [10,11,12,13]. In a short period of time, several cases of TMA after a cardiovascular surgery were diagnosed at our institution. We, therefore, conducted a retrospective study to evaluate potential risk factors associated to the occurrence of TMA in patients with severe CSA-AKI, the impact on in-hospital renal outcome and mortality.

## 2. Materials and Methods

We performed a retrospective, single-center analysis over a 15-month period between January 2018 and March 2019 among patients admitted to the intensive care unit (ICU) of the Department of Cardiac Surgery at the University Hospital, LMU Munich. The study protocol was approved by the local ethics committees of the LMU Munich (Project number 19-370, date of approval 12 June 2019).

### 2.1. Subjects

We included all patients requiring postoperative intensive care monitoring and renal replacement therapy (RRT) due to AKI within one week after cardiovascular surgery (from day 0 to day 7) during the observation period. All patients were aged ≥18 years old. Patients with ESKD with need for chronic hemodialysis prior to surgery were excluded from the study. RRT due to AKI was initiated according to the judgement of the attending physician and/or intensivist specialist using the AKI criteria of the KDIGO guidelines for AKI in patients with abrupt renal impairment after surgery [14,15,16].

All patients with severe CSA-AKI needing RRT were included in the study and categorized as “TMA” or “Non-TMA” for comparative analysis. Patients with severe CSA-AKI without signs of TMA were designated as “Non-TMA”. Patients fulfilling TMA criteria upon severe CSA-AKI were reported as “TMA”. TMA diagnosis was defined as laboratory findings indicating (1) Coombs-negative MAHA (hemoglobin ≤10g/dL, elevated LDH >250 U/L, haptoglobin <0.3 g/L, and/or bilirubin >1.2 mg/dL, and/or the presence of schistocytes ≥1 ‰, and a negative direct antiglobulin test or Coombs test); (2) non-immune thrombocytopenia (platelet count <150 G/L and a negative heparin-induced thrombocytopenia (HIT) test); (3) post-surgery severe AKI needing RRT; (4) and/or signs of neurological impairment (i.e., drowsiness, delayed awakening following anesthesia or postoperative delirium).

Other systemic disorders with a more plausible explanation for the new onset of signs associated with MAHA, thrombocytopenia and/or AKI (e.g., major bleeding, severe sepsis, disseminated intravascular coagulation (DIC), acute liver failure, use of veno-arterial extracorporeal membrane circulation (v-a ECMO), prosthetic valve-associated hemolysis) were excluded. Subsequently, differential diagnosis of other TMA forms and possible triggers (such as malignancy, autoimmune disease, pregnancy, triggering medications, bone marrow transplantation, vitamin B12 deficiency) were excluded based on past medical history, current medications and clinical and laboratory evaluation. Thrombotic thrombocytopenic purpura (TTP) was ruled out by measuring ADAMTS13 activity. ADAMTS13 activity >10% excluded TTP as a differential diagnosis. STEC-HUS was excluded by lack of diarrhea and/or negative stool microbiology. Overall assessment was conducted in interdisciplinary case discussions by an experienced team of physicians from the department of nephrology, anesthesiology and cardiothoracic surgery.

Of note: current classification systems do not report surgery as a triggering event for aHUS, therefore, we used the term TMA instead of aHUS in the current paper. Proof of activation of the alternative complement system or genetic evaluation was not possible in all patients due to the retrospective design of the study.

### 2.2. Kidney Function and Laboratory Parameters

Kidney function was assessed by serum creatinine levels (mg/dL), estimated glomerular filtration rate (eGFR, mL/min) and serum urea (mg/dL). eGFR was calculated using the CKD-EPI (Chronic Kidney Disease Epidemiology Collaboration) formula.

Data on kidney function parameters prior to surgery were extracted from the data base of the hospital laboratory or from external analyses as documented in medical reports shortly before surgery. Postoperative kidney function was evaluated using maximum creatinine and urea levels between day 0 and day 7 postoperatively. As RRT was started in all patients, eGFR was assumed to be 0 mL/min. Furthermore, values of creatinine, eGFR and urea were obtained at time of discharge.

Routine laboratory parameters were obtained from the database of the Department for Laboratory Medicine, including ADAMTS13 activity and parameters of complement activation. Coombs test was routinely performed at the Department of Transfusion Medicine, Cellular Therapy and Hemostaseology of our hospital.

### 2.3. Data Analysis and Endpoints

Patient data, including external and internal medical records, surgery reports, anesthesiology reports, and clinical evolution notes, were obtained from the electronic clinical records of the Department of Cardiac Surgery of our hospital using the electronic information system “KAS” (Klinischer Arbeitsplatz) and “LAMP”. Statistical analyses were performed using STATA, version 16 IC (Stata Corp LLC, TX, USA) and Microsoft Excel, version Microsoft Office 365 (Microsoft Corporation, Redmond, WA, U.S.). Continuous variables were assessed for normality using histograms and Shapiro–Wilk test. Due to skewness of the data, median and interquartile range (IQR) were used as measures of central tendency and dispersion, respectively. Non-parametric tests, i.e., Wilcoxon rank sum (Mann–Whitney) test and Wilcoxon signed rank tests, were used for comparison between and within groups.

Categorical data (i.e., dichotomous variables) were expressed as absolute and relative frequencies. Odds ratio was used to assess relative risks in dichotomous variables. Pearson’s chi-squared test and two-tailed Fischer’s exact test were used to compare dichotomous variables. We considered *p*-values ≤ 0.05 as statistically significant.

The primary outcome was the association of risk factors (baseline demographic and clinical characteristics, type of surgery, and intraoperative management) with TMA occurrence in patients requiring RRT upon CSA-AKI. Secondary outcomes included in-hospital renal outcome after TMA occurrence based on surrogate parameters for kidney function and need for RRT, in-hospital and ICU length of stay, and overall mortality.

## 3. Results

A total of 893 patients were admitted to the ICU after a cardiovascular surgery between January 2018 and March 2019. Out of those, 69 patients (7.7%) needed RRT due to severe AKI occurring within one week after surgery (i.e., from day 0 to day 7). Among patients with CSA-AKI requiring RRT, 15 patients (21.7%) fulfilled TMA criteria. The study cohort is depicted in Figure 1.

### 3.1. Baseline Characteristics

Demographic characteristics of the study population at baseline prior to surgery are depicted in Table 1. Overall, median age was 68 (IQR: 62–76) years with females representing only 39.1% among the total patient cohort. However, females were overrepresented in the TMA group as compared to the non-TMA group (9/15, 60.0%, vs. 18/44, 33.3%, *p* = 0.063), although this did not reach statistical significance. TMA patients were slightly, but not significantly, younger than patients without a TMA (66 (IQR: 45–72) vs. 70 (IQR: 62–76) years, *p* = 0.112). Overall, cardiovascular risk factors, coronary artery disease and impaired left ventricular ejection fraction were frequent in both groups without significant differences. Nevertheless, aortic aneurysms were significantly more frequent in the TMA group than in the non-TMA group (46.7% vs. 14.8%, *p* = 0.014). There were no differences regarding preoperative routine baseline laboratory parameters such as complete blood count or coagulation parameters (see Table 2).

### 3.2. Baseline Kidney Function

Whereas almost half of the non-TMA group (48.2%) presented with impaired kidney function (eGFR ≤ 60 mL/min) prior to surgery, only 20% of TMA patients had chronic kidney disease before surgery (Table 1). Consecutively, eGFR at baseline was significantly higher in the TMA group (92 (IQR: 57–100) vs. 60.5 (IQR: 44–79) mL/min, *p* = 0.004). Table 2 shows surrogate kidney function parameters including creatinine, eGFR and urea of all patients at baseline.

### 3.3. Surgery Characteristics

Aortic replacement with or without associated aortic valve surgery was overall the most commonly performed surgical intervention with 26 out of 69 patients (37.7%). Compared to non-TMA patients, aortic surgery was significantly more frequently performed among patients with a postoperative TMA (60.0% vs. 31.5%, OR 3.26, 95% CI 1.0013–10.64). Further intraoperative characteristics are shown in Table 3. No significant differences were observed regarding the CPB time (230 (IQR: 156–272) vs. 190 (IQR: 145–243) min, *p* = 0.299) or in the use of hypothermic circulatory arrest (HCA) between the groups (7 (46.7%) vs. 17 (31.5%), OR 1.9, 95% CI 0.59–6.11). Nevertheless, HCA time was significantly shorter in TMA patients (26 (IQR: 22–35) vs. 59 (IQR: 54–75) min, *p* = 0.003). Moreover, cross clamp-time was longer in patients with TMA than in non-TMA patients (161 (IQR: 109–181) vs. 121 (IQR: 74.5–144) min, *p* = 0.048); however, use of aortic cross-clamping did not differ between groups (13 (86.7%) vs. 43 (79.6%), OR 1.6, 95% CI 0.33–8.48). No significant differences were observed regarding the lowest core body temperature. Interestingly, use of veno-arterial extracorporeal membrane circulation (ECMO) was less frequent in TMA patients when compared to non-TMA patients (13.3% vs. 40.7%, OR 0.22, 95% CI 0.05–1.09), although this difference was not significant.

### 3.4. Postsurgical Outcomes

#### 3.4.1. Thrombotic Microangiopathy

From all patients with severe CSA-AKI, 15 patients fulfilled TMA criteria within one week (from day 0 to day 7) after surgery. In addition to severe AKI, 10 TMA patients presented with neurological impairment post-surgery, including drowsiness, delayed awakening following anesthesia and delirium. 

The laboratory parameters of subjects fulfilling TMA criteria after surgery are displayed in Table 4 (for comparison to the non-TMA group see Appendix A). TMA patients presented with profound anemia (hemoglobin 7.8 (IQR: 7.2–8) g/dL), thrombocytopenia (40 (IQR: 28–45) G/L), reduced haptoglobin (0.07 G/L), elevated LDH levels (1777 (IQR: 859–2631) U/L) and elevated schistocyte count (9 (IQR: 6–20) ‰) after surgery. Coagulation parameters including fibrinogen, INR and D-Dimer were only modestly disturbed (Table 4). Heparin-induced thrombocytopenia type II (HIT type II), disseminated intravascular coagulation (DIC), and a positive coombs test were excluded in all TMA patients.

Infections, as a potential trigger of TMA, were excluded by appropriate testing. None of the patients presented with diarrhea prior to surgery, making the diagnosis of STEC-HUS unlikely. All administered drugs were not described in the literature as potential TMA triggering factors, except for one patient with heart transplantation in 2004 and stable tacrolimus dose. The median ADAMTS13 activity was 64% (IQR: 57–77), excluding TTP as a differential diagnosis. ADAMTS13 was not measured in three out of the 15 patients, however the clinical course made a diagnosis of TTP unlikely. Complement diagnostics, including activity of the classical and alternative pathway, were performed in 10 patients. Nine of those showed increased complement activation. Upon exclusion of TTP and STEC-HUS, aHUS was proposed as final diagnosis for all patients with clinical features of TMA.

#### 3.4.2. TMA Treatment

Patients with a TMA diagnosis but without spontaneous improvement were initiated on therapeutic plasma exchange (TPE). Out of the 10 patients treated with TPE, four did not show improvement and were treated with eculizumab after obtaining the ADAMTS13 results. Vaccines against meningococcus (Meningococcal conjugate vaccine ACWY and/or serogroup B meningococcal vaccine) were administered prior to eculizumab treatment. Antibiotic prophylaxis with oral penicillin (ciprofloxacin in case of penicillin allergy) was concurrently initiated and continued during the course of eculizumab treatment, accounting for the complete in-hospital stay in these patients. All patients treated with eculizumab showed increased complement activation and normal ADAMTS13 activity.

#### 3.4.3. Renal Outcome and Mortality

Indications for RRT after surgery were not different between both groups, including drop in urine output (oliguria/anuria), elevated kidney function parameters, electrolyte disbalances or metabolic acidosis. In-hospital need for RRT was significantly longer among TMA patients than in the control patients with AKI (20 (IQR: 14–30) vs. 7.5 (IQR: 2–25) days, *p* = 0.019).

Death censored kidney function parameters on discharge revealed no significant differences between patients with TMA after surgery and patients without TMA, as measured by serum creatinine (2.3 (IQR: 1.0–4.0) vs. 1.55 (IQR: 1.2–2.7) mg/dL, *p* = 0.322), eGFR (32 (IQR: 0–66) vs. 45 (IQR: 0–67) mL/min, *p* = 0.743) and urea (80 (IQR: 40–111) vs. 78 (35–108) mg/dL, *p* = 0.885). Overall, loss of kidney function was significant within both groups (Appendix A). However, loss of kidney function was more pronounced in patients with TMA than in patients without a TMA needing RRT upon AKI (ΔeGFR −54 (IQR −8–−81) vs. −17 (IQR: −2–−80) mL/min, *p* = 0.062) (Appendix A). Further kidney function parameters on discharge are shown in Appendix A. Lastly, the number of patients discharged on hemodialysis was comparable between both groups (26.7% vs. 33.3%, *p* = 0.73).

In-hospital length of stay was overall 24 (IQR: 12–40) days and no significant differences were found between both groups (Appendix A). In addition, ICU length of stay was similar among the groups (12 (IQR: 10–27) vs. 17 (IQR: 7–33) days, *p* = 0.575). No in-hospital deaths were accounted in the TMA group, whereas 24 deaths were reported among patients requiring RRT upon AKI without a TMA (0% vs. 44.4%, *p* = 0.001).

## 4. Discussion

AKI is a severe complication following cardiovascular surgery and an independent risk factor for mortality. Multiple injuring mechanisms have been suggested to induce CSA-AKI, with some reports suggesting TMA as a postsurgical complication leading to AKI [1,9]. TMAs are very rare diseases which, if untreated, can result in irreversible ESKD and death [7]. In the present study, we identified TMA cases in a cohort of patients with severe AKI needing RRT after cardiovascular surgery and compared them to patients without TMA. We assessed for potential risk factors that might be associated with postsurgical TMA in the context of CSA-AKI, the in-hospital renal outcome and mortality.

In the 15-month period, 7.7% of all patients undergoing cardiovascular surgery in our university hospital needed RRT within the first week due to AKI. Among these patients, 15 were diagnosed with a postoperative TMA, representing 21.7% of patients with severe AKI and 1.7% of all treated patients on the ICU after a cardiovascular procedure. Consistent with the current literature, post-surgical TMA was more frequent among female patients and was diagnosed within the first week after surgery in our patient cohort [6,9].

Past medical history regarding cardiovascular risk factors and other comorbidities were overall similar between groups. However, aortic aneurysms were significantly more frequent in the TMA group than in the non-TMA group. Accordingly, aortic surgeries with or without aortic valve replacement were associated with the occurrence of TMA after a cardiovascular procedure. Interestingly, postoperative TMA in our cohort was not associated with the use of veno-arterial ECMO or longer CPB time intraoperatively, different from the reported influence of both factors on the occurrence of AKI [17,18]. Furthermore, our results show that TMA patients required longer cross-clamp time and shorter HCA time intraoperatively, while the use of these techniques themselves were not significantly different between groups. This is consistent with the literature, where current studies have not found HCA time to influence the incidence of AKI after cardiovascular surgeries [18,19]. However, our results should be interpreted with caution due to the small sample size of the subgroups.

Several case reports and case series have described the occurrence of TMA after cardiovascular surgeries as well as surgeries in general [9,10,20]. However, current classification systems do not include surgeries as triggers of TMA. In primary aHUS, the dysregulation of the complement system leads to the clinical picture of TMA. In trigger-induced or secondary aHUS (also known as secondary TMA) a two-step model is proposed in which an endogenous predisposition (e.g., genetic predisposition) and a triggering condition (e.g., pregnancy, solid organ transplantation, drugs) are required for aHUS manifestation [21]. Based on our study, cardiovascular surgery should be considered a triggering event for aHUS, which is underlined by the detection of complement activation in our TMA patient cohort. The pathophysiological mechanism of postsurgical TMA is not well defined. Extensive endothelial damage, especially after aortic surgery, could lead to overactivation of the alternative complement pathway. Complex cardiovascular surgeries could either represent a severe triggering condition not necessarily requiring a genetic predisposition for TMA manifestation, or could unmask mild complement abnormalities [6,21]. Further studies are necessary to evaluate underlying pathomechanisms (e.g., genetic abnormalities or the presence of CFH autoantibodies) in these patients [22].

Out of the 15 patients with TMA, 5 attained clinical remission spontaneously. Ten patients were treated with TPE and 6 of them achieved clinical remission after several days. The remaining four TMA patients required eculizumab as resolution of hemolysis and clinical recovery were not achieved with TPE alone. Use of TPE in the initial therapy of aHUS is accepted when TTP still remains a differential diagnosis and delay in treatment could imply a higher risk for the patients. Furthermore, TPE could help normalize uncontrolled complement activation as denoted in one study, where TPE induced remission in 55–80% of patients with complement-mediated aHUS [8]. However, no randomized trials are available for TPE in aHUS. Similar to our cohort, a previous literature review of 65 patients with postsurgical TMA reported use of TPE in 83% of patients with clinical improvement in 81% of them. About 26% of patients receiving TPE needed additional therapy, consistent with our results [9].

After exclusion of TTP, complement blockade with eculizumab is highly recommended in complement-mediated aHUS, whereas its use in secondary aHUS/TMA forms remains controversial [6]. Before eculizumab, prognosis of aHUS was very poor leading in 50% of the cases to ESKD and a mortality rate of 5% during the first episode [23]. Several studies have proven the efficacy of this therapy and demonstrated a positive influence on renal outcome [8,24]. Eculizumab was administered in four TMA patients of our cohort not responding to TPE as recommended for aHUS and continued after discharge with good laboratory and clinical response. There is no consensus regarding the optimal duration of the therapy with eculizumab, but known triggering conditions and the result of genetic testing of complement abnormalities should be incorporated in this decision. Based on our findings, eculizumab should be promptly started upon high suspicion of aHUS in patients not responding to the initial therapy and discontinuation should be carefully evaluated by the treating physician considering genetic testing. In view of the postulated two-step model, prolonged treatment with eculizumab might not be necessary upon resolution of the triggering condition.

Finally, our data suggest an important role of TMA in patients with severe CSA-AKI regarding short-term renal outcome. Remarkably, preoperative kidney function in these patients was significantly better compared to patients without a TMA after surgery. However, TMA resulted in a significant loss of eGFR (approximately 50 mL/min) despite therapy, accounting for more than half of the baseline renal function. Follow-up studies are essential for determining long-term renal outcomes and quality of life in these patients. Despite significant loss in kidney function suggesting severe TMA, none of the patients with post-surgery TMA died while hospitalized. In contrast, death occurred in 24 out of 54 (44.4%) patients requiring RRT upon CSA-AKI without TMA.

It is important to bear in mind that post-surgical TMA is a diagnosis of exclusion and its identification can be challenging. Physicians should consider TMA as possible cause for AKI in patients undergoing cardiovascular surgeries (especially involving the aorta) who present with signs of persistent MAHA (Coombs-negative hemolysis), thrombocytopenia, and organ damage (e.g., AKI, neurological impairment), after excluding other more common causes in this setting (e.g., sepsis, DIC, major bleeding, mechanical-induced hemolysis by ECMO, HIT). Careful laboratory assessment and clinical suspicion is crucial for early diagnosis of TMA upon cardiovascular surgery.

There are several limitations to this study. First, the retrospective observational study design limits the completeness of data, the inclusion of genetic analysis and the long-term follow-up. Furthermore, generalizability of the data might be limited due to the small sample size. However, we present the largest study on TMA associated with cardiovascular surgery reported so far. Some degree of selection bias may overestimate the prevalence of TMA after cardiovascular surgeries, as this was a single-center study at a university hospital with a higher number of complex cardiovascular procedures. Additionally, the rather short observation period further reduced the sample size, however, routine assessment of MAHA, ADAMTS13 activity and complement activation was not implemented before the study period. Prospective, multi-center studies are essential for further evaluation of potential triggers and optimal treatment strategies. The information provided by our series may be helpful for the design of future prospective studies on this matter.

In conclusion, TMA is an important differential diagnosis in patients with severe CSA-AKI. The results of our observational study suggest in particular complex aortic surgeries as potential trigger for the development of TMA. Early diagnosis and timely treatment including plasma exchange and eculizumab resulted in favorable renal and patient outcome. Even so, TMA occurrence had a profound impact on short-term renal function with significant loss of GFR. Only prospective interventional studies and long-term follow-up can establish optimal strategies for diagnosis and therapeutic interventions.

## Figures and Tables

**Figure 1 jcm-09-02900-f001:**
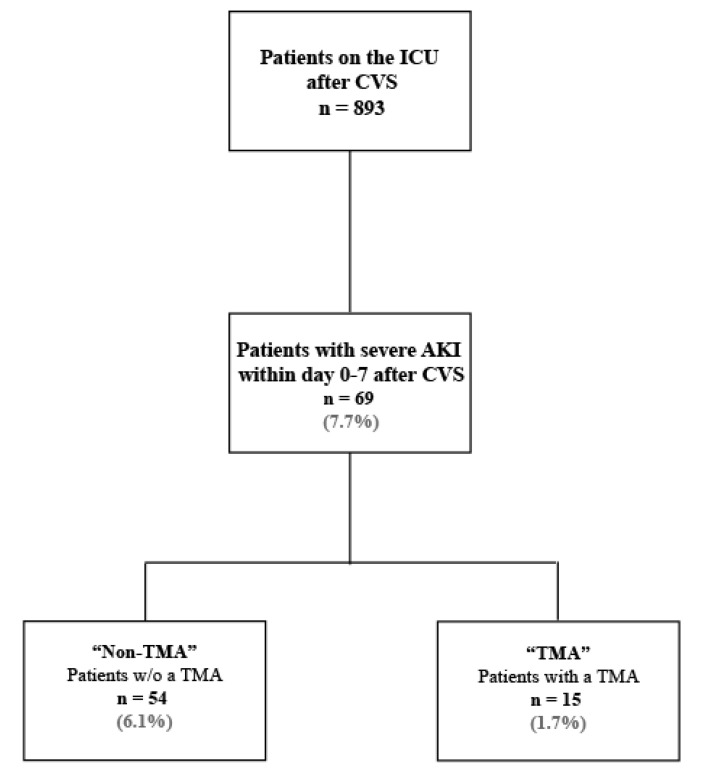
Patients admitted to the CVS unit between January 2018 and March 2019 after a cardiovascular procedure. AKI: acute kidney injury, CVS: cardiovascular surgery, ESKD: end-stage kidney disease, RRT: renal replacement therapy, TMA: thrombotic microangiopathy, w/o: without. N = number of patients, (%) percent out of total number of patients admitted to the ICU after CVS (N = 893).

**Table 1 jcm-09-02900-t001:** Baseline characteristics.

	TMA (*n* = 15)		Non-TMA (*n* = 54)		*p*-Value
Baseline characteristics	Median (IQR)		Median (IQR)		
Female (n, %)	9	(60)	18	(33.3)	0.063
Age (years)	66	(45–72)	70	(62–76)	0.112
Body-mass index (kg/m^2^)	23.2	(20.8–30.4)	25.8	(22.4–28.9)	0.531
Height (m)	1.68	(1.63–1.75)	1.75	(1.65–1.85)	0.098
Weight (kg)	76	(63–85)	78	(69–94)	0.284
Diagnosis on admission	*n* (%)		*n* (%)		*p*-value
Aortic valve disease					
Aortic stenosis	1	(6.7)	10	(18.5)	0.434
Aortic insufficiency	8	(53.3)	19	(35.2)	0.240
Aortic endocarditis	2	(13.3)	6	(11.1)	1.000
Mitral valve disease					
Mitral stenosis	0	(0.0)	5	(9.3)	0.578
Mitral insufficiency	5	(33.3)	23	(42.6)	0.568
Mitral endocarditis	0	(0.0)	5	(9.3)	0.578
Tricuspid valve disease					
Tricuspid insufficiency	3	(20.0)	14	(25.9)	0.746
Tricuspid stenosis	0	(0.0)	0	(0.0)	NA
Aortic disease					
Aortic aneurysm	7	(46.7)	8	(14.8)	0.014
Aortic dissection Type A	1	(6.7)	9	(16.7)	0.442
Coronary artery disease (CAD)	7	(46.7)	34	(57.4)	0.561
Reduced LVEF	4	(26.7)	27	(42.6)	0.373
End stage heart failure	0	(0.0)	11	(16.7)	0.189
Previous heart surgery	7	(46.7)	19	(33.3)	0.375
	**TMA (*n* = 15)**		**Non-TMA (*n* = 54)**		***p*-Value**
Cardiovascular history & risk factors	*n* (%)		*n* (%)		
Diabetes mellitus (Type 1 or 2)	2	(13.3)	12	(22.2)	0.718
Arterial hypertension	10	(66.7)	38	(70.4)	0.761
Hypercholesterinemia	10	(66.7)	27	(50.0)	0.381
Family history for CAD	6	(40.0)	6	(14.8)	0.063
Obesity	4	(26.7)	17	(31.5)	1.000
History of smoking	6	(40.0)	23	(42.6)	1.000
Macroangiopathy (cerebral or peripheral)	10	(66.7)	28	(51.9)	0.386
Renal history	*n* (%)		*n* (%)		*p*-value
Chronic kidney disease (eGFR≤ 60ml/min)	3	(20.0)	26	(48.2)	0.076
Other pathologies	*n* (%)		*n* (%)		*p*-value
Autoimmune disease	3	(20.0)	3	(5.7)	0.112
History of malignancy	1	(6.7)	9	(16.7)	0.442
Medication	*n* (%)		*n* (%)		*p*-value
ACE-Inhibitors/AT1-Blockers	8	(53.3)	31	(57.4)	0.778
Beta blockers	9	(60.0)	38	(70.3)	0.535
Statins	8	(53.3)	24	(44.4)	0.572
Aspirin/Clopidogrel	7	(46.7)	24	(44.4)	1.000
Ticlopidine	0	(0.0)	1	(1.9)	1.000
OAC (Marcumar/NOACs)	6	(40.0)	25	(46.3)	0.773
Immunosuppression	2	(13.3)	2	(3.7)	0.204

TMA: Thrombotic microangiopathy, kg: kilograms, m: meters, kg/m^2^: kilograms per square meters, eGFR: estimated glomerular filtration rate, CAD: coronary artery disease, LVEF: left ventricular ejection fraction, ACE-Inhibitors: angiotensin-converting enzyme inhibitors; AT1-Blockers: angiotensin II type 1 receptor blockers, NOACs: novel oral anticoagulants, OAC: oral anticoagulants, NA: not applicable.

**Table 2 jcm-09-02900-t002:** Preoperative laboratory parameters.

	TMA (*n* = 15)		Non-TMA (*n* = 54)		*p*-Value
	Median (IQR)		Median (IQR)		
Bilirubin (mg/dL)	0.7	(0.4–0.9)	0.9	(0.6–1.4)	0.075
LDH (U/L)	314	(202–388)	326	(226–445)	0.755
Hemoglobin (g/dL)	12.3	(11.2–14.1)	12.7	(10.4–13.9)	0.788
Thrombocytes (G/L)	189	(152–250)	202	(147–251)	0.884
INR	1.1	(1.0–1.2)	1.1	(1.0–1.4)	0.142
**Kidney function parameters**					
Creatinine (mg/dL)	0.9	(0.8–1.3)	1.3	(1.0–1.7)	0.005
eGFR (mL/min)	92	(57–100)	60.5	(44–79)	0.004
Urea (mg/dL)	38	(30–48)	55	(39–79)	0.025

LDH: lactate dehydrogenase, INR: international normalized ratio, eGFR: estimated glomerular filtration rate calculated with CKD-EPI formula.

**Table 3 jcm-09-02900-t003:** Surgery characteristics.

	TMA (*n* = 15)		Non-TMA (*n* = 54)			OR	95% CI
Surgery type *	*n* (%)		*n* (%)				
Aortic valve repair	3	(20.0)	13	(24.1)	0.78		(0.19–3.23)
w/o aortic replacement							
Mitral valve	3	(20.0)	15	(27.8)	0.65		(0.16–2.63)
Tricuspid valve	3	(20.0)	11	(20.4)	0.97		(0.23–4.08)
Aortic replacement with or w/o aortic valve procedure	9	(60.0)	17	(31.5)	3.26		(1.0013–10.64)
Aortocoronary bypass	3	(20.0)	13	(24.1)	0.79		(0.19–3.23)
Heart transplantation	0	(0.0)	4	(7.4)	0.00		NA
Postoperative interventions							
Postoperative v-a ECMO	2	(13.3)	22	(40.7)	0.22		(0.05–1.09)
Multiple surgeries (2 or more)	9	(60.0)	31	(57.4)	1.11		(0.34–3.56)
Intraoperative characteristics							
HCA	7	(46.7)	17	(31.5)	1.9		(0.59–6.11)
Aortic cross-clamping	13	(86.7)	43	(79.6)	1.6		(0.33–8.48)
	Median (IQR)		Median (IQR)		*p*-value		
CPB time (min)	230	(156–272)	190	(145–243)	0.299		
Cross-clamp time (min) ^#^	161	(109–181)	121	(74.5–144)	0.048		
HCA time (min) ^Δ^	26	(22–35)	59	(54–75)	0.003		
SACP time (min) ^Δ^	22	(21–35)	59	(44–75)	0.005		
Lowest core temperature (°C)	32	(25–35)	33.5	(25.5–35.5)	0.413		
SACP temperature (°C) ^Δ^	22	(22–23)	22	(22–22.3)	0.820		

All displayed valve procedures include valve replacement and/or repair. Abbreviations: v-a ECMO: veno-arterial extracorporeal membrane circulation. CPB: cardiopulmonary bypass. HCA: hypothermic circulatory arrest. SACP: selective antegrade cerebral perfusion. w/o: without. NA: not applicable. * Some patients underwent multiple procedures in one surgery. ^Δ^ Only patients undergoing HCA intraoperatively were included in this analysis. ^#^ Only patients undergoing aortic cross-clamping intraoperatively were included in this analysis.

**Table 4 jcm-09-02900-t004:** Laboratory parameters of patients with TMA (*n* = 15).

	Preoperative		Postoperative (Day 0–7)		*p*-Value	Normal Range
	Median (IQR)		Median (IQR)			
Bilirubin (mg/dL)	0.7	(0.4–0.9)	3.6	(2.9–5.7)	0.002	<1.2
LDH (U/L)	314	(202–388)	1777	(859–2631)	0.005	<249
Hemoglobin (g/dL)	12.3	(11.2–14.1)	7.8	(7.2–8.0)	0.0007	11.5–15.4
Thrombocytes (G/L)	189	(152–250)	40	(28–45)	0.0007	176–391
Haptoglobin (g/L)	n.d.		0.07	(0.07–0.07)	NA	0.3–2.0
INR	1.1	(1.0–1.2)	1.2	(1.1–1.5)	0.409	0.8–1.2
Fibrinogen (mg/dL)	n.d.		419	(338–590)	NA	210–400
D-Dimer (ug/dL)	n.d.		8	(5.4–12.2)	NA	<0.5
Schistocyte count (‰)	n.d.		9	(6–20)	NA	<1

LDH: lactate dehydrogenase, INR: international normalized ratio, n.d. not determined, NA: not applicable.

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
