# Peer review of "Severe Acute Kidney Injury in Cardiovascular Surgery: Thrombotic Microangiopathy as a Differential Diagnosis to Ischemia Reperfusion Injury. A Retrospective Study"

_jcm, 2020, doi:10.3390/jcm9092900_

Round 1

Reviewer 1 Report

No further comments.

Reviewer 2 Report

The authors addressed all concerns. The revised manuscript looks good and suitable for publication

This manuscript is a resubmission of an earlier submission. The following is a list of the peer review reports and author responses from that submission.

Round 1

Reviewer 1 Report

This is a study submitted by Grigorescu et al reporting on THROMBOIC MICROANGIOPATHY (TMA) as an uncommon yet important cause of acute kidney injury after cardiovascular surgery (CVS). The reports included a total of 893 patients who required ICU admission after CVS, 79 patient required renal replacement therapy (RRT), and only 15 patients fulfilled the criteria for TMA. The authors reported that patients with RRT due to TMA where mostly females with relatively younger age and better kidney function prior to surgery - compared to control group (patients who required RRT NOT due to TMA). Authors reported no significant difference regarding renal outcomes or in hospital length of stay between TMA and control group. Interestingly, in hospital death rate was significantly higher in the control group. The authors attribute the occurrence of TMA after CVS to possibly ischemia reperfusion injury, and cardiopulmonary bypass injury. The report is very clinically relevant to practicing nephrologists, well written and appropriately designed. The small size cohort is the main limitation to the study which was appropriately addressed by the authors. The tables are very informative and detailed. I would accept this nice paper for publication with no major edits.

Author Response

Response to Reviewer 1 Comments

Point 1: This is a study submitted by Grigorescu et al reporting on THROMBOIC MICROANGIOPATHY (TMA) as an uncommon yet important cause of acute kidney injury after cardiovascular surgery (CVS). The reports included a total of 893 patients who required ICU admission after CVS, 79 patient required renal replacement therapy (RRT), and only 15 patients fulfilled the criteria for TMA. The authors reported that patients with RRT due to TMA where mostly females with relatively younger age and better kidney function prior to surgery - compared to control group (patients who required RRT NOT due to TMA). Authors reported no significant difference regarding renal outcomes or in hospital length of stay between TMA and control group. Interestingly, in hospital death rate was significantly higher in the control group. The authors attribute the occurrence of TMA after CVS to possibly ischemia reperfusion injury, and cardiopulmonary bypass injury. The report is very clinically relevant to practicing nephrologists, well written and appropriately designed. The small size cohort is the main limitation to the study which was appropriately addressed by the authors. The tables are very informative and detailed. I would accept this nice paper for publication with no major edits. 

Response 1:

We thank the reviewer for the kind comment. Due to the request of reviewer 2 we changed some aspects of the paper and hope that reviewer 1 is still satisfied with the current version of the paper.

Reviewer 2 Report

Fundamentally, this study reports the comparative analysis between TMA and non-TMA patients. Does this suffice as a full length research article? I highly doubt it. Unless there is specific research question investigated by the researchers here, this manuscript is not fit for publication.

Here are some of my comments:
1. Classification of TMA is not reported here. The authors only analyzed ADAMTS13 activity to exclude one type. What about others? Please elaborate on the different types of TMA exhibited by the patient group. Clearly, these TMA patients cannot be clubbed in one general TMA category.

2. Was the data normally distributed across the three groups? It is confusing if there are three groups or two groups? Figure 1 shows three groups, but analysis shows two. Please report normality test results and data is not normally distributed then non-parametric test results should be reported.

3. Throughout the document, the authors have used period and comma interchangeably. This is not expected from an official manuscript submission. Fig 2 – Seems like the authors added these plots as an afterthought. Poor contribution from the authors and clear oversight by senior authors as well. Furthermore, what do these plots convey? There are no axes, labels, or ticks on these plots. There is no clear message.

Furthermore, can the authors tell any significant differences between any of the time points? It clearly shows that whatever intervention (or cases group) was administered to these patients, there was no differences in creatinine, eGFR, or urea levels. Authors cannot claim any effect of “treatment”, unless there is a two-way ANOVA analysis for the three outcomes variables with time and group as factors.

Author Response

Response to Reviewer 2 Comments

Point 1: Fundamentally, this study reports the comparative analysis between TMA and non-TMA patients. Does this suffice as a full length research article? I highly doubt it. Unless there is specific research question investigated by the researchers here, this manuscript is not fit for publication.

Response 1:

We thank the reviewer for the valuable comments. In our paper we describe an uncommon yet important cause of acute kidney injury after cardiovascular surgery (CVS). Thrombotic microangiopathies (TMA) are medical emergencies with potentially high morbidity and mortality and a substantial risk of chronic kidney damage. Retrospective studies are valuable instruments to describe new observations and are the basis for further prospective studies. We conducted a retrospective study to evaluate potential risk factors associated to the occurrence of TMA in patients with severe CSA-AKI, the impact on in-hospital renal outcome and mortality. We concretized our research question as follows: “The primary outcome was the association of risk factors (baseline demographic and clinical characteristics, type of surgery and intraoperative management) with TMA occurrence in patients requiring RRT after CVS. Secondary outcomes included in-hospital renal outcome after TMA occurrence based on surrogate parameters for kidney function and need for RRT, in-hospital and ICU length of stay and overall mortality.” (see 2.3 Data analysis and endpoints)

Point 2: Here are some of my comments:

  1. Classification of TMA is not reported here. The authors only analyzed ADAMTS13 activity to exclude one type. What about others? Please elaborate on the different types of TMA exhibited by the patient group. Clearly, these TMA patients cannot be clubbed in one general TMA category.

Response 2:

We can assure the reviewer that all patients were treated by an experienced team of clical physicians familial with the diagnosis and treatment of all forms of TMA. TTP and STEC-HUS were excluded. For aHUS triggering factors were extensively evaluated and excluded as possible. However, we could not find any of the known triggering factors. Furthermore, the lack of sympotms prior to surgery as well as initiation of the TMA episode with the cardiovascular surgery points to CVS as a so far not recognized triggering factor for activation of the complement system and TMA. It is within the retrospective design of our study that proof of activation of the alternative complement system or genetic evaluation was not possible in all patients.

We extended the text es follows: “The diagnosis and treatment of TMA were determined by an experienced team of clinical physicians and in interdisciplinary case discussions, excluding other systemic disorders associated with MAHA and thrombocytopenia. Differential diagnosis of TMA forms and other possible TMA triggers (such as malignancy, autoimmune disease, pregnancy, triggering medications, bone marrow transplantation, vitamin B12 deficiency) were excluded based on past medical history, current medications and clinical and laboratory evaluation. Thrombotic thrombocytopenic purpura (TTP) was ruled out by measuring ADAMTS13 activity. STEC-HUS was excluded by lack of diarrhea and/or stool microbiology.  Current classification systems do not report surgery as a triggering event for TMA, therefore we use the term TMA instead of aHUS in the current paper. Proof of activation of the alternative complement system or genetic evaluation was not possible in all patients due to the retrospective design of the study.” (see 2.1 Subjects)

Point 3: 2. Was the data normally distributed across the three groups? It is confusing if there are three groups or two groups? Figure 1 shows three groups, but analysis shows two. Please report normality test results and data is not normally distributed then non-parametric test results should be reported.

Response 3:

We thank the reviewer for the comment and advice. “Continuous variables were assessed for normality using histograms and Shapiro-Wilk test. Due to skewness of the data, median and interquartile range (IQR) were used as measures of central tendency and dispersion, respectively. Non-parametric tests, i.e. Wilcoxon Rank Sum (Man Whitney) test and Wilcoxon Signed Rank tests, were used for comparison between and within groups.” (see 2.3 Data analysis and endpoints) Consecutively, data were changed within the text and tables. Furthermore, we changed Figure 1 to make it clear that we compared patients with and without TMA requiring RRT after CVS. We also removed the combined numbers of both groups from the tables now shoing only both groups in comparison. In summary, the main messages have not changed with the new data analysis.

Point 4: 3. Throughout the document, the authors have used period and comma interchangeably. This is not expected from an official manuscript submission. Fig 2 – Seems like the authors added these plots as an afterthought. Poor contribution from the authors and clear oversight by senior authors as well. Furthermore, what do these plots convey? There are no axes, labels, or ticks on these plots. There is no clear message.

Response 4:

We apologize that we have overlooked a few commata as we changed from comma to period. We removed the original Figure 2 and inserted supplementary Figure 2 to make the “In-hospital evolution of kidney function – baseline vs. discharge” more visible between Controls and TMA patients.

Point 5: Furthermore, can the authors tell any significant differences between any of the time points? It clearly shows that whatever intervention (or cases group) was administered to these patients, there was no differences in creatinine, eGFR, or urea levels. Authors cannot claim any effect of “treatment”, unless there is a two-way ANOVA analysis for the three outcomes variables with time and group as factors... 

Response 5:

This was a retrospective analysis. It is the main message to describe the occurrence of TMA after CVS and aortic surgery as major risk factor. We further point out that the TMA espisode had a major impact on loss of kidney function. Of course the treatment effect (with PE or eculizumab) can only be acknowleged in a well designed prospective study. However, from other studies with aHUS the negative impact on renal and patient survival without specific treatment is well known, and for “complement-mediated aHUS” only treatment with eculizumab has dramatically changed this outcome in recent years. We therefore treated all TMA patients after interdisciplinary discussion with an accepted standard of care. With this approach outcome for these TMA patients was favourable but not perfect, when looking at the loss of kidney function. We agree that we have no control group without treamtent.

We thank the reviewer for the valuable comment. However, we don’t think that ANOVA analysis is useful in our study setting. TMA patients without spontaneous resolution within a few days were treated with PE, and those without resolution with PE received eculizumab. We therefore slightly changed our conclusion, removing any specific therapeutic recommendations. “In conclusion, TMAs are an important differential diagnosis in patients with severe CSA-AKI. Identification of risk factors, early diagnosis and treatment are essential for renal and patient survival. Long-term follow-up and prospective studies are needed.”
